# TATA: Stance Detection via Topic-Agnostic and Topic-Aware Embeddings

**Hans W. A. Hanley**
Stanford University
hhanley@cs.stanford.edu

**Zakir Durumeric**
Stanford University
zakir@cs.stanford.edu

## Abstract

Stance detection is important for understanding different attitudes and beliefs on the Internet. However, given that a passage's stance toward a given topic is often highly dependent on that topic, building a stance detection model that generalizes to unseen topics is difficult. In this work, we propose using contrastive learning as well as an unlabeled dataset of news articles that cover a variety of different topics to train topic-agnostic/TAG and topic-aware/TAW embeddings for use in downstream stance detection. Combining these embeddings in our full TATA model, we achieve state-of-the-art performance across several public stance detection datasets (0.771 $F_1$-score on the Zero-shot VAST dataset). We release our code and data at https://github.com/hanshanley/tata.

## 1 Introduction

Stance detection is the task of determining the attitude of a piece of text with regard to a particular topic or target. Stance detection is often utilized to gauge public opinion toward products, political issues, or public figures. For example, the statement *"I think Abraham Lincoln would make a good President"* would have a *Pro* stance concerning the topic of *Abraham Lincoln*. Simultaneously, the same text would be *Neutral* to the topic of *Fruits*.

Several works have built topic-specific stance classifiers that have achieved promising results for particular targets (Mohtarami et al., 2018; Xu et al., 2018); however, these same approaches have often failed to generalize to topics outside of their training domain, limiting their real-world use. Recent works have sought to alleviate this issue by utilizing general stance features (Liang et al., 2022a), common sense reasoning (Liu et al., 2021a), and generalized topic representations (Allaway and McKeown, 2020, 2023; Liang et al., 2022b), achieving better performance in predicting stance on previously unseen topics (*Zero-shot* stance detection)

and on topics for which there are few training examples (*Few-shot* stance detection). In this work, building on these insights, we create and release synthetic datasets for designing topic-aware/**TAW** embeddings and generalized topic-agnostic/**TAG** stance embeddings for use in downstream stance classification tasks. Taking advantage of *both* the topic-aware and topic-agnostic perspectives and embeddings, our full **TATA** model achieves state-of-the-art performance in *both* the Zero-shot and Few-shot stance classification settings on the VAST benchmark dataset (Allaway and McKeown, 2020).

Specifically, utilizing an unstructured news article dataset (Hanley and Durumeric, 2023) and extracting topics from within particular passages using the T5-Flan language model (Chung et al., 2022), we train a topic-aware/TAW embedding layer with a triplet loss (Hermans et al., 2017). By using triplet loss, we force training elements with similar topics to have similar embeddings and training elements with different topics to have dissimilar embeddings. By using an unlabelled dataset of news articles with a diverse and wide set of automatically extracted topics when creating these topic embeddings, rather than simply relying on the set of topics with expensive and carefully labeled stance datasets as in other works (Allaway and McKeown, 2020; Liang et al., 2022b), enables these topic embeddings to incorporate relationships between many different topics.

Similarly, for our topic-agnostic/TAG embeddings, after first extending the original VAST training dataset with paraphrased versions of the original texts, we utilize contrastive learning objectives to then extract generalized stance features (Saunshi et al., 2019). By using contrastive learning, we force training instances with the same stance to have similar embeddings while simultaneously forcing training examples with different stances to have different embeddings. This enables our TAG layer to model general stance features that

are important for classifying the stance in a given example.

Combining both the topic-aware/TAW and topic-agnostic/TAG perspectives into one model TATA, we achieve state-of-the-art performance on the VAST dataset in both Zero-shot ($0.771$ $F_1$-score compared to $0.723$ in prior work (Liang et al., 2022b)) and Few-shot ($0.741$ $F_1$-score compared to $0.715$ in prior work) settings. Our TATA model further achieves competitive results on the SEM16t6 dataset (Mohammad et al., 2016) in the Zero-shot setting. By training on synthetic datasets, taking advantage of stance features that are common across different topics/TAG, *and* deriving specific features particular to given topics utilizing a rich embedding latent/TAW, our TATA model thus achieves state-of-the-art performance on current benchmarks for stance classification.

## 2 Related Work and Background

**Stance Detetection.** Biber and Finegan define stance as "the expression of a speaker's attitude, standpoint, and judgment" toward a topic (1988). In the most common implementation, stance detection tasks consist of inputting a passage-topic pair (where the passage is a longer text and the topic is a noun phrase) and outputting the stance from among $\{Pro, Against, Neutral\}$ of that passage toward that given topic. Stance detection, in addition to understanding political attitudes (Darwish et al., 2020), has been utilized to improve fact checking (Dulhanty et al., 2019; Si et al., 2021; Bekoulis et al., 2021), and to gauge general public opinion (Alturayeif et al., 2023).

Given its real-world applications, several works have sought to improve upon domain-specific and baseline stance-detection methods to identify stances in *Few-shot* settings (where the classifier is evaluated on a large number of topics for which it had few training examples) and *Zero-shot* scenarios (where the classifier is evaluated on new, previously seen topics) (Liang et al., 2022a; Allaway and McKeown, 2020, 2023). For example, in addition to developing one of the most varied benchmark datasets for evaluating Zero-shot stance detection VAST, Allaway and McKeown (2020) also pioneered the use of generalized topic representations to perform Zero-shot and Few-Shot stance detection. More recently, Zhu et al. (2022), improving upon prior work, utilized targeted outside knowledge to add information for stance detection

on previously unseen topics. Li and Yuan (2022), utilized generative models (*e.g.*, GPT2) to create synthetic data to engineer training data for unseen topics. Finally, Liang et al. (2022b) further proposed target-aware graph contrastive learning to learn relationships between topic-based representations for downstream stance classification.

**Contrastive Learning.** Contrastive learning, which seeks to build representations by differentiating similarly and differently labeled inputs, has become an increasingly popular version of self-supervised learning (Liu et al., 2021b; Rethmeier and Augenstein, 2023; Gao et al., 2021). Contrastive learning is often utilized to build robust representations that can then be utilized in downstream tasks (Gunel et al., 2020) including generating sentence embeddings (Gao et al., 2021; Wu et al., 2022), image classification (Park et al., 2020; Wang and Qi, 2022), and image captioning (Dai and Lin, 2017). As in our work, contrastive learning has also increasingly been utilized to model stance features (Liang et al., 2022a,b; Mohtarami et al., 2019). For example, Liang et al. (2022b) utilized in-batch contrastive learning to differentiate between examples of stance classes while also building a graph of prototypical topics to train topic-specific features for downstream stance detection.

## 3 Topic-Aware/TAW Dataset

As previously stated, within this work, we seek to build informative topic-aware/TAW embeddings that can then be utilized in downstream stance detection. As such to pre-train these embeddings, we collect a new dataset that includes a wide range of topics. This new dataset, designed for creating topic-aware embeddings, consists of sets of quadruplets such that $D_{Taw} = \{x_i = (p_i, t_{i1}, t_{i2}, q_i)\}_{i=1}^{N}$ where $p_i$ is a passage, $t_{i1}$ is $p_i$'s corresponding topic, $t_{i2}$ is a paraphrase of the topic $t_{i1}$, and $q_i$ is a semantically similar passage to $p_i$.

### 3.1 Passage Selection and Noisy Topic Identification

To first build a dataset of passages that contain wide-ranging multifaceted topics, we make use of a dataset from Hanley and Durumeric (2023) of 15.90 million news articles collected between January 1, 2022, and May 1, 2023, from 3,074 news sites. From these articles, we randomly subselect a set of 100,000 articles (with at most 1000 articles from any given single website). For each article in our new set, we take the first paragraph (up to

100 words) for inclusion as passages within our TAW dataset. In order to use this set of passages to pre-train a topic-aware encoding layer, as in Allaway and McKeown (2020), we subsequently identify the topics present within these passages. As in Allaway and McKeown (2020), for each passage $p_i$ we define a candidate topic as a noun phrase in the passage from the constituency parse generated using the Python Spacy library.[1] If no noun phrases are identified, we remove the passage from our dataset and randomly select another article and passage for use in our dataset. Unlike Allaway and McKeown (2020), who then heuristically select one of the noun phrases as a topic, once candidate noun phrases have been identified, we subsequently utilize the off-the-shelf T5-Flan-XL (Chung et al., 2022) model from Huggingface[2] to further refine our topic identification. The particular model that we utilize was fine-tuned to identify topics within passages using the following prompt: `Select the topic that this about:{text}\n\n{options}\n\n,{answer}` and we also utilize this prompt to identify the topic of our passages.

To ensure a diversity of different topics within our dataset, we allow at most three passages within our dataset to have the same topics. We again remove passages-topic pairs that did not fit this criterion, creating another passage-topic pair using Hanley and Durumeric (2023)'s dataset if necessary. Altogether, our TAW training dataset contained 97,984 unique topic noun phrases, illustrating the diversity of different topics considered. After identifying 100,000 passage-topic pairs for use in our training dataset, we identify an additional 10,000 passage-topic pairs, each with a unique topic and with nonoverlapping topics with our TAW training dataset, as our validation set. We note that while we only utilize 110,000 pairs here, this TAW dataset could easily be extended to utilize more passages and topics.

## 3.2 Topically Similar Passages

Once we identified the topics of our set of passages, we subsequently identified other passages within Hanley and Durumeric (2023)'s dataset that were similar/about the same topic as our set of passages. To do so, we use an off-the-shelf version of the MPNet (Song et al., 2020) large language model (LLM) fine-tuned on semantic similarity tasks.[3] Embedding all constituent 100-word passages from Hanley and Durumeric (2023)'s dataset and indexing them with the `FAISS` library (Johnson et al., 2019), a library for efficient semantic similarity search, for each passage $p_i$ in our dataset, we subsequently identify the passage $q_i$ that had the highest semantic similarity to the given passage $p_i$. To ensure that each topically similar passage was stylistically different from its similar passage $p_i$, we only include passage pairs if they originated from two different websites. We set the minimum similarity of two passages to be topically similar at a cosine similarity threshold of 0.70 (Hanley et al., 2023; Grootendorst, 2022). Across our final dataset, each topically identified similar passage $q_i$ had an average/median cosine similarity of 0.796/0.786 to its assigned passage $p_i$ in our training dataset and an average/median similarity of 0.786/0.764 in our validation set.

We further augment this dataset with different paraphrases of the extracted topic strings. By paraphrasing these topics, we model the different ways of expressing the same topic. To make these paraphrases, we utilize another publicly available T5-based paraphraser `Parrot` that was trained on short texts (Damodaran, 2021). For more details on this paraphraser, see Appendix B and for example topic paraphrases see Appendix D. We note that some of the topics are particular named entities such as locations, person names, *etc...*, and thus cannot be effectively paraphrased. As such, we utilize the `Spacy` named entity recognizer to identify instances of topics that cannot be effectively paraphrased. Altogether, we remove entities in the following classes {'PERSON', 'GPE', 'LOC', 'TIME', 'PERCENT', 'QUANTITY', 'ORDINAL', 'MONEY', 'DATE'}.

Our final dataset $D_{Taw} = \{x_i = (p_i, t_{i1}, t_{i2}, q_i)\}_{i=1}^{N}$ consists of 110,000 quadruplets, where the passages $p_i$ were taken from news articles (Hanley and Durumeric, 2023), the topics $t_{i1}$ were identified by T5-FLan XL, $t_{i2}$ are `Parrot` paraphrases of the $t_{i1}$ topics, and the semantically similar passages $q_i$ were identified using MPNet. With permission from Hanley and Durumeric (2023), we release these quadruplets and the splits upon request to a form at `https://github.com/hanshanley/tata`.

---

[1] `https://spacy.io/`
[2] `https://huggingface.co/google/flan-t5-xl`

[3] `https://huggingface.co/sentence-transformers/all-mpnet-base-v2`

## 4 Augmented VAST Dataset

In addition to developing our own dataset for training TAW embeddings, we further extend Allaway and McKeown (2020)'s VAST dataset to pre-train our topic-agnostic/TAG embeddings. As previously noted, the VAST dataset is one of the most varied stance datasets. VAST consists of a dataset $D_{Vast} = \{x_i = (p_i, t_i, s_i)\}_{i=1}^N$, where $p_i$ a passage, $t_i$ the topic of the passage, and $s_i$ is passage $p_i$'s stance toward the topic $t_i$ with $s_i \in \{Pro, Neutral, Against\}$. We augment the training dataset with paraphrases of the original passages *and* topics while pretraining our TAG embedding layer enabling the generation of additional cases of passages with the exact stances and lexically similar topics to those in the VAST dataset.

To create paraphrases of the original VAST dataset's passages, we rely on the pre-trained `Dipper` paraphraser[4], a T5-based model fine-tuned to paraphrase paragraph-level texts (Krishna et al., 2023). See Appendix C for details on the hyperparameters we utilized for `Dipper` and see Appendix E for example `Dipper` paraphrases. By utilizing the T5-based `Dipper` model, we augment the original VAST dataset without changing the original meaning of the original VAST passages. As with our TAW dataset, we further augment VAST with different paraphrases of the original topic strings with the same methodology outlined in Section 3.2. By paraphrasing these topics, we again model the different ways of expressing the same topic.

Altogether, by paraphrasing each row from the VAST dataset (paraphrasing individual texts as many as 16 times) and by extracting out as many as 10 paraphrases of each topic phrase/word within the original VAST dataset $D_{VAST}$, we extend the original training dataset of 13,477 examples to a total of 743,644 different examples $D_{VAST_{aug}}$.

## 5 Methods

We develop a model (TATA) that combines topic-aware and topic-agnostic embedding layers to perform Zero and Few-shot stance detection. This model (Figure 1) consists of the topic-aware embeddings layer (5.1), the topic-agnostic embedding layer (5.2), two attention layers using the output of the topic-aware embedding layer and the topic-agnostic embedding layer (5.3), and finally a two-

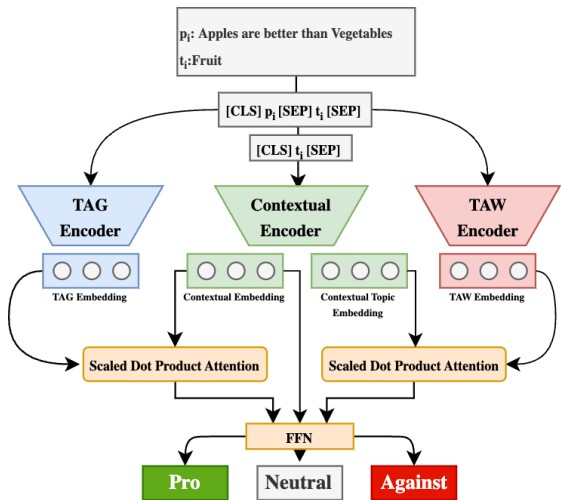

Figure 1: TATA Model.

layer feed-forward neural network for stance classification (5.4).

### 5.1 Topic-Aware/TAW Embedding Layer

As previously noted, we train a topic-aware embedding layer to extract topic-aware features for use in downstream stance classification. As in Allaway and McKeown (2020), we seek to design topic representations that can be used to learn topic-specific stance features in downstream tasks. However, unlike previous work (Liang et al., 2022a,b; Allaway and McKeown, 2020), we do not use representative topic clusters as embeddings of particular topics during training. Instead, we use our TAW dataset and contrastive learning techniques to develop a rich general latent representation of different topics. Once trained, this topic latent, which implicitly contains the relationship between topics, can then be utilized to get informative generalized topic representations. After training this layer, we freeze this layer for use in the rest of our TATA model.

Specifically, we utilize a triplet loss to learn similar representations for similar topics and to differentiate different topics. This is such that we embed each example $x_i \in D_{Taw} = \{x_i = (p_i, t_{i1}, q_i, t_{i2})\}_{i=1}^N$ using a contextual word model by inputting by $[CLS]p_i[SEP]t_{i1}[SEP]$ and $[CLS]q_i[SEP]t_{i2}[SEP]$ and outputting the hidden states of the $[CLS]$ tokens $\mathbf{h}_{p_i}$ and $\mathbf{h}_{q_i}$. Then using the hidden states of the $[CLS]$ tokens, we minimize with a triplet loss the Euclidean distance between the two hidden $\mathbf{h}_{p_i}, \mathbf{h}_{q_j}$ that share a common paraphrased topic. We condition the hidden state $\mathbf{h}_{q_j}$ using a paraphrased version $t_{i2}$ of the topic $t_{i1}$ to guard against our model only learning to recognize moving together sentences with

---

[4]https://huggingface.co/kalpeshk2011/dipper-paraphraser-xxl

the exact same topic $t$. Putting this together, we train our embedding layer such that given the outputted set of hidden vector pairs $\{\mathbf{h}_{p_i}, \mathbf{h}_{q_j}\}_{i,j=0}^{N_b}$ in a batch $\mathcal{B}$ of size $N_b$, we treat all pairs where $i = j$ as positive pairs and pairs where $i \neq j$ as negative pairs. Within each batch $\mathcal{B}$, the triplet loss is computed across all positive pairs in the batch:

$$L_{taw} = \frac{1}{N_b} \sum_{\mathbf{h}_{p_i}, \mathbf{h}_{q_j} \in \mathcal{B}, i=j} l^{taw}(\mathbf{h}_{p_i}, \mathbf{h}_{q_j})$$

$$l^{taw}(\mathbf{h}_{p_i}, \mathbf{h}_{q_j}) = \sum_{i=j, i \neq k} [\max(0, ||\mathbf{h}_{p_i} - \mathbf{h}_{q_j}||_2^2$$
$$-||\mathbf{h}_{p_i} - \mathbf{h}_{q_k}||_2^2)]$$

We use a triplet loss for our TAW layer rather than a contrastive loss because normal contrastive loss pushes similarly labeled elements to have a near-zero distance between them (Xuan et al., 2020). However, for our TAW encoding layer, given the noisiness of our $D_{Taw}$ dataset, while we wish for examples with similar topics to have similar embeddings, here we do not wish to train them to have the same representation. Triplet loss only ensures that negative examples are further away in the latent space than positive examples. Namely, triplet loss enables us to ensure that our noisy pairs have more similar TAW representations between themselves than with other examples in our dataset.

## 5.2 Topic-Agnostic/TAG Embedding Layer

As noted by Allaway and McKeown (2023), embeddings that encode general stance features have been shown to improve a model's ability to discern the stance of different passages. As in Liang et al. (2022b), we thus use contrastive learning to differentiate the $\{Pro, Neutral, Against\}$ stance classes and build a TAG embedding layer with our augmented VAST dataset $D_{VAST_{aug}}$. After training this layer, we freeze it for use in the rest of our TATA model.

We train our TAG layer such that we embed each example $x_i \in D_{VAST_{aug}} = \{x_i = (p_i, t_i, s_i)\}_{i=1}^N$ using a contextual word model, inputting $[CLS]p_i[SEP]t_i[SEP]$ and outputting the hidden vector of the [CLS] token for each $x_i$. Then, given a set of hidden vectors $\{\mathbf{h}_i\}_{i=0}^{N_b}$, where $N_b$ is the size of the batch, we perform contrastive learning with that batch. This is such that for each batch $\mathcal{B}$, for an *anchor* hidden embedding $\mathbf{h_i}$ within the batch, the set of hidden vectors $\mathbf{h_i}, \mathbf{h_j} \in \mathcal{B}$ vectors where $i \neq j$ are considered a positive pair if their corresponding stances $s_i, s_j$ are equivalent; other

pairs where $s_i \neq s_j$ are considered negative pairs. Within each batch $\mathcal{B}$, the contrastive loss is computed across all positive pairs in the batch such that:

$$L_{tag} = -\frac{1}{N_b} \sum_{\mathbf{h}_i \in \mathcal{B}} l^{tag}(\mathbf{h}_i)$$

$$l^{tag}(\mathbf{h}_i) = \log \frac{\sum_{j \in \mathcal{B} \setminus i} \mathbb{1}_{[s_i=s_j]} \exp(\frac{\mathbf{h}_i^\top \mathbf{h}_j}{\tau ||\mathbf{h}_i|| ||\mathbf{h}_j||})}{\sum_{j \in \mathcal{B} \setminus i} \exp(\frac{\mathbf{h}_i^\top \mathbf{h}_j}{\tau ||\mathbf{h}_i|| ||\mathbf{h}_j||})}$$

where $\tau$ is the temperature parameter.

## 5.3 TATA Attention

Finally, once our topic-agnostic/TAG and topic-aware/TAW layers are fully trained, as shown in Figure 1, we utilize them as embedding layers within our full TATA model. Specifically, taking as input instances $x_i$ such that $\{x_i = (p_i, t_i, s_i)\}$ our TATA model, uses the outputted hidden vectors $\mathbf{h}_{taw}$ and $\mathbf{h}_{tag}$ in attention layers with contextual embeddings of the passage and topic. This is such that after getting the output of the TAW layer $\mathbf{h}_{taw}$ and our TAG layer $\mathbf{h}_{tag}$, and outputting another contextual embedding representation $\mathbf{h}_{passage|topic}$ of the joint input of the passage-topic pair, and a contextual embedding $\mathbf{h}_{topic}$ the topic $t_i$, we finally utilize a scaled dot product attention (Vaswani et al., 2017) between $\mathbf{h}_{taw}$ and $\mathbf{h}_{topic}$ as well between $\mathbf{h}_{tag}$ and $\mathbf{h}_{passage|topic}$:

$$r_{topic} = \sum_i a_i \mathbf{h}_{taw}^{(i)},$$
$$a_i = \text{softmax}\left(\lambda \mathbf{h}_{taw}^{(i)} \cdot (W_{taw} \mathbf{h}_{topic})\right)$$

$$r_{stance} = \sum_i b_i \mathbf{h}_{passage|topic}^{(i)},$$
$$b_i = \text{softmax}\left(\lambda \mathbf{h}_{passage|topic}^{(i)} \cdot (W_{tag} \mathbf{h}_{tag})\right)$$

where $W_{tag} \in \mathbb{R}^{E \times E}$ and $W_{taw} \in \mathbb{R}^{E \times E}$ are learned parameters, and $\lambda = 1/\sqrt{E}$. This enables us to obtain a representation $r_{topic}$ that captures the relationship between the generated TAW embedding and our topic; in this way, we manage to better determine which aspect of the general TAW latent space representation of the topic is important in representing the topic for downstream stance classification. We further get the representation $r_{stance}$ that captures how much the calculated stance features extracted by our TAG embedding apply to our given passage.

| | Train | Validation | Test |
|---|---|---|---|
| # Examples | 13,477 | 2,062 | 3,006 |
| # Unique passages | 1,845 | 682 | 786 |
| # Zero-shot topics | 4,003 | 383 | 600 |
| # Few-shot topics | 638 | 114 | 159 |

Table 1: Data statistics for the VAST dataset.

## 5.4 Label Prediction

Once $r_{topic}$ and $r_{stance}$ are calculated, we concatenate them with $\mathbf{h}_{passage|topic}$ using a residual connection. We then feed the resulting representation into a two-layer feed-forward network with softmax activation to compute the output probabilities for $\{Pro, Neutral, Against\}$. We minimize cross-entropy loss while training.

## 6 Experiments

### 6.1 Pre-training

**TAW Training Data.** As pre-training data for our TAW embedding layer, we utilize our TAW $D_{Taw}$ dataset of passage pairs with shared topics. As previously stated, this dataset includes 100,000 passage pairs with shared topics as training data and 10,000 passage pairs as validation data.

**TAG Training Data.** As pre-training data for our TAG embedding layer, we utilize our augmented VAST dataset $D_{VAST_{aug}}$ consisting of 743,644 unique passage-topic-stance triplets. We utilize the validation split of the original VAST dataset as validation data.

**Pre-training Details.** We utilize DeBERTa-v3-base as our contextual model and to jointly embed our passage-topic pairs, where our passage-topic pairs are mapped to a 768-dimensional embedding using the outputted hidden state of the $[CLS]$ token (He et al., 2022). We use DeBERTa given its larger vocabulary (128,100 tokens vs. 30,522 tokens in BERT (Devlin et al., 2019)) and its better overall documented performance on various downstream tasks (He et al., 2022). While encoding the passage-topic pairs, we use a maximum length of 512 possible tokens as opposed to the first 200 tokens as in other works (Allaway and McKeown, 2020). Unlike in previous works (Allaway and McKeown, 2020; Liang et al., 2022b), we further do not remove stopwords and punctuation from our datasets, electing to use the full unprocessed version of original texts and their topics.

While pretraining the TAG and TAW layers, we set the learning rate to $1 \times 10^{-5}$ and use AdamW as the optimizer (Kingma and Ba, 2015). Due to computational constraints, while pretraining, we use a batch size of 16. While pre-training and using a contrastive loss, we set the temperature parameter $\tau$ to 0.07 as in Liang et al. (2022b). For our TAG layer, we ended training after 2 epochs, and for our TAW layer, we ended training after 1 epoch. We completed all training on an Nvidia A6000 GPU. Once trained, we freeze the TAG and TAW embedding layers.

### 6.2 Training

**Training Data.** While much of our pre-training data was artificially generated or was unlabeled data gathered from news websites, after we freeze our TAG and TAW layers, for training our TATA stance classifier, we use only the original VAST dataset. As previously noted, this dataset consists of 13,477 passage-topic-stance triplets. We give an overview of this dataset in Table 1.

**TATA Training Details.** We use similar settings training as we did for pre-training except for changing to a batch size of 32. In our feed-forward layers, we utilize dropout with $p = 0.30$. While training, we utilize early stopping with a patience of 3.

### 6.3 Testing

**Testing Data.** We utilize the testing split within the VAST dataset. Following prior work (Liang et al., 2022b; Allaway and McKeown, 2020; Liu et al., 2021a), when reporting results on this dataset, we calculate the macro-averaged $F_1$ for each stance label. We present results separately for the Few-shot (where the detector is evaluated on a large number of topics for which it had few training examples) and Zero-shot (where the detector is evaluated on new, previously seen topics) settings.

**Baseline and Comparison Models.** We compare our proposed TATA model with several other baselines including TGA-Net (Allaway and McKeown, 2020), TOAD (Allaway et al., 2021), CKE-Net (Liu et al., 2021a), BERT-GCN (Lin et al., 2021), and JointCL (Liang et al., 2022b). As an additional baseline, we further train a model based on DeBERTa-v3-base that takes the hidden state of the $[CLS]$ for the combined passage-topic input followed by a two-layer feed-forward neural network to predict the stance. Finally, to understand the respective importance of the TAW and the TAG embedding layers, we further train a TAG architecture that takes our original TATA architecture and removes the TAW embedding layer as well as a

| | Zero-Shot VAST | | | | Few-Shot VAST | | | |
|---|---|---|---|---|---|---|---|---|
| | Pro | Against | Neutral | All | Pro | Against | Neutral | All |
| BERT | 0.546 | 0.584 | 0.853 | 0.661 | 0.554 | 0.597 | 0.796 | 0.646 |
| TGA-Net | 0.554 | 0.585 | 0.858 | 0.665 | 0.589 | 0.595 | 0.805 | 0.663 |
| CKE-Net | 0.612 | 0.612 | 0.880 | 0.702 | 0.644 | 0.622 | 0.835 | 0.701 |
| BERT-GCN | 0.583 | 0.606 | 0.869 | 0.686 | 0.628 | 0.643 | 0.830 | 0.697 |
| TOAD | 0.426 | 0.367 | 0.438 | 0.410 | - | - | - | - |
| JointCL | 0.649 | 0.632 | 0.889 | 0.723 | 0.632 | 0.667 | 0.846 | 0.715 |
| DeBERTa | 0.680 | 0.683 | 0.900 | 0.755 | 0.659 | 0.657 | 0.869 | 0.728 |
| TAW | 0.672 | 0.709 | 0.903 | 0.760 | 0.656 | 0.677 | 0.869 | 0.736 |
| TAG | 0.681 | 0.687 | 0.901 | 0.756 | **0.665** | 0.655 | 0.868 | 0.729 |
| TATA | **0.695** | **0.711** | **0.905** | **0.771** | **0.665** | **0.683** | **0.873** | **0.741** |

Table 2: $F_1$-scores of models on benchmarks from the VAST dataset. We bold the highest/best score in each column. We obtain scores for BERT and TGA-Net from Allaway and McKeown (2020), scores for CKE-Net and BERT-GCN from Liu et al. (2021a), and scores for TOAD and JointCL from Liang et al. (2022b).

## 7 Experimental Results

Our models achieve a noticeable increase in performance over prior work. We present our results on the VAST test dataset in Table 2. We observe that by using the unedited text, including stopwords, using a max token length of 512, and by finetuning a DeBERTa model, we achieve similar results to JointCL (Liang et al., 2022b). This illustrates that utilizing a model with a larger token vocabulary (128,100 in DeBERTa vs. 30,522 in BERT) can largely assist with the stance detection task. However, we further observe that by utilizing the TAG and TAW features in our full TATA models, we can increase our performance, achieving a 0.771 $F_1$ score in the Zero-shot setting and a 0.741 $F_1$ score in the Few-shot setting.

**Ablation Studies.** We perform an ablation study in order to understand the relative importance of our TAG and TAW layers. As seen in Table 2, our TAW model performs better than our TAG model in both Zero-shot and Few-shot scenarios suggesting the importance of the TAW topic latent in predicting stance. By removing this layer from our TATA model, our model's score in the Zero-shot and Few-shot scenarios decreases by 0.015 $F_1$ and 0.012 $F_1$ respectively.

Given the reliance of our TAW model on the informative TAW topic latent for predicting stance, we perform another study to investigate whether it is simply predicting the stance of input texts using only the input topic text. We do this primarily to ensure that our TAW model is actually utilizing the input texts and is not relying primarily on the topic embeddings. To do so, for our TAW embedding layer rather than passing $[CLS]p_i[SEP]t_i[SEP]$ to the layer as input, we pass in $[CLS][SEP]t_i[SEP]$. As seen in Table 3, predicting based on the topic significantly hurts our results in the Zero-shot and Few-shot settings, with our model nearly always predicting the *Neutral* class. This behavior largely conforms to the definition of the *Neutral* class; without any input text, the stance of an empty string to a given topic should be neutral. This suggests that our TAW model is not only picking up on the topic but is also, as expected, heavily considering the text when predicting the stance.

Examining our TAG model, we only observe slight improvements over our DeBERTa model. Indeed, plotting the t-SNE (Van der Maaten and Hinton, 2008) of the entire VAST validation dataset, as seen in Figure 2, while we do observe the formation of clear stance clusters during training on the VAST validation set, these clusters are not exact. Despite this, we note that incorporating the TAG layer embeddings to our TATA model *did* lead to better downstream stance classification (an increase of 0.011 $F_1$ and 0.005 $F_1$ in the Zero-shot and Few-shot settings respectively). This reinforces prior work (Allaway and McKeown, 2023; Liang et al., 2022b) that has found that general stance features, incorporated with other features and knowledge, can lead to better downstream stance classification.

### 7.1 Error Analysis

**Challenging Linguistic Phenomena.** As in Allaway and McKeown (2020), we analyze the perfor-

TAW architecture that removes the TAG embedding layer. Lastly, we note to improve the consistency and robustness of our results, we train each model with a different random seed a total of five times and report the average of those five different runs in our results.

|        | Zero-Shot VAST | | | | Few-Shot VAST | | | |
|--------|-----|---------|---------|-------|-----|---------|---------|-------|
| Model  | Pro | Against | Neutral | All   | Pro | Against | Neutral | All   |
| TAW    | 0.034 | 0.000 | 0.519 | 0.184 | 0.020 | 0.000 | 0.507 | 0.176 |

Table 3: Performing an ablation study testing whether our TAW model utilizes only the topic for determining the stance of a given text, we find that our model *does* utilize the passage input, with this reduced model nearly always predicting the *Neutral* class.

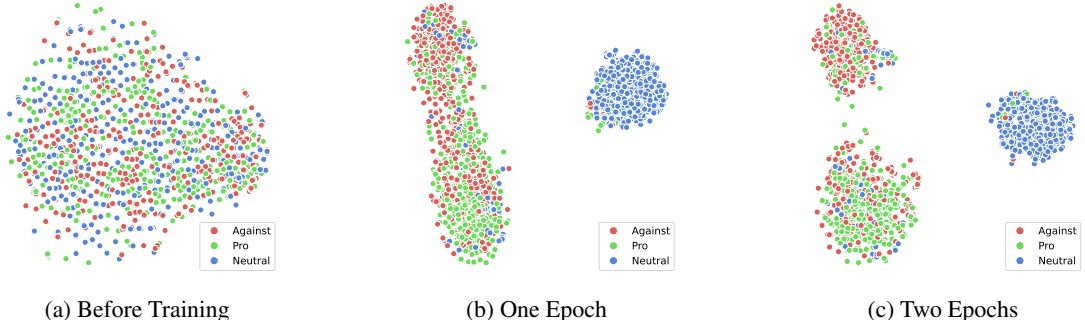

(a) Before Training     (b) One Epoch     (c) Two Epochs

Figure 2: As we train the topic-agnostic/TAG layer of our model on our augmented VAST training set, while not separating perfectly (illustrating the need for additional features) clear *Pro*, *Against*, and *Neutral* clusters appear in the t-SNE of the embeddings of the VAST validation dataset. As confirmed elsewhere (Allaway and McKeown, 2020; Liang et al., 2022a), the *Neutral* category of examples is the most differentiable from *Pro* and *Against* categories.

| Model | Qte N=539 | Sarc N=431 | Imp N=1231 | mlS N=952 | mlT N=1802 |
|-------|-----------|------------|------------|-----------|------------|
| TGA-Net | 0.661 | 0.637 | 0.623 | 0.547 | 0.624 |
| BERT    | 0.625 | 0.587 | 0.600 | 0.541 | 0.610 |
| DeBERTa | 0.709 | 0.713 | 0.667 | 0.578 | 0.676 |
| TAW     | 0.712 | 0.722 | 0.673 | **0.620** | 0.689 |
| TAG     | 0.710 | **0.731** | 0.668 | 0.568 | 0.676 |
| TATA    | **0.714** | 0.730 | **0.693** | 0.603 | **0.703** |

Table 4: Accuracies on various challenging linguistic phenomena in the VAST test set. We give accuracies to directly compare our results against Allaway and McKeown (2020). We utilize scores for BERT and TGA-Net from Allaway and McKeown (2020).

mance of our TATA network on five challenging linguistic phenomena provided within the VAST test dataset. The phenomena within the VAST dataset for passage-topic pairs are as follows: (1) *Qte*: the passage contains quotations, (2) *Sarc*: the passage contains sarcasm, (3) *Imp*: the topic is not expressly contained in the passage, (4) *mlS*: the passage is among a set with multiple topics with different non-neutral stance labels, and (5) *mlT*: the passage has multiple topics. As seen across the various phenomena, our set of models outperforms the TGA-Net and BERT baselines from Allaway and McKeown (2020), with TATA having the best overall performance. We find that our TATA model performs particularly better in cases where there are quotations ($Qte$), where there are multiple topics ($mlT$), and where the passage does not expressively contain the topic ($Imp$). This again suggests that our

TATA model is able to effectively utilize the TAW informative latent space of topics. We note that our TAW model, also performs second-best in each of these scenarios, besting our TATA model in performance on cases where individual passages have topics with different non-neutral stances ($mlS$). In these ($mlS$) cases, the model must learn to not simply always predict the same stance for a given text; and as seen in Table 4, while our topic-ware models perform better in these scenarios compared to all other models, across every model tested, they perform the worst in on this particular challenging linguistic phenomenon. We leave to future work to explore how to encourage models to better differentiate stances in passages with conflicting stances on multiple topics.

**Performance on Topics Included in TAW dataset.**
We note that several of the Zero-shot topics in the VAST dataset had lexically similar topics in our TAW dataset. We thus determine if the number of lexically similar topics included in the TAW dataset correlated with our TATA model's performance on those topics. In this way, we test if learning the specific tested topics within our latent space was important in later predicting a stance involving the topic. To identify lexically similar topics, as in Allaway and McKeown (2020), we represent each topic in the Zero-shot test set using pre-trained GloVe word embeddings (Pennington et al., 2014) and determine the number of topics examples in our TAW dataset with cosine similarity to these

| Target | Pro | Against | Neutral |
|---|---|---|---|
| Donald Trump | 148 | 200 | 260 |
| Hillary Clinton | 163 | 565 | 256 |
| Feminist Movement | 268 | 511 | 170 |
| Legalization of Abortion | 167 | 544 | 222 |
| Atheism | 124 | 464 | 145 |
| Climate Change | 335 | 26 | 203 |

Table 5: Data statistics for the SEM16t6 dataset.

topics greater than 0.90. We do not find a statistically significant correlation between the number of lexically similar topics in our TAW dataset and whether our TATA model predicted the corresponding VAST test instance's stance correctly (p-value =0.353). This illustrates that our TAW model's latent was not particularly biased toward the specific topics that we included within our TAW dataset.

**Performance in Few-shot vs. Zero-Shot Settings.** Given our model's greater performance in the Zero-shot setting than the Few-shot setting, we measure whether the number of lexically similar training examples with regards to a given topic within VAST (*e.g.,* where a lexically similar topic is as previously defined in our paper and Allaway and McKeown (2020) as having a GloVe cosine similarity above 0.9) is correlated with whether our system correctly labels a given instance of that topic. If there is a positive correlation, this may indicate that the topics with few examples may be noisy/biasing our results (as those with no similar training examples at all have higher $F_1$-scores than those in the Few-shot setting). In the Few-shot setting, we find a small Pearson correlation (0.186, p-value =0.021) between the number of semantic similar training examples and the percentage of a given topic's test examples that are labeled correctly by our TATA model (0.2675, p-value=0.001 for the TAG model; 0.199, p-value=0.012 for the TAW model; 0.215, p-value= 0.007 for the DeBERTa model). Together with the other models (Allaway and McKeown, 2020; Liu et al., 2021a; Liang et al., 2022b) that perform worse in the Few-shot setting, this suggests some noisiness and a tendency of our model to be biased on topics in VAST with only one or two training examples. As reported by Allaway and McKeown (2020), there was high (75%) but not perfect agreement among labelers of the original VAST, further helping to explain this phenomenon.

**Generalizability.** As was seen in Table 2, our models have better performance in Zero-shot and Few-shot settings than other baselines on the VAST dataset. However, to further confirm these re-

| | Zero-Shot SEM16t6 | | | | | | |
|---|---|---|---|---|---|---|---|
| | DT | LA | HC | FM | CC | A | Avg |
| JointCL | 0.505 | 0.495 | 0.548 | 0.538 | 0.397 | 0.545 | 0.505 |
| TOAD | 0.495 | 0.462 | 0.512 | 0.541 | 0.309 | 0.461 | 0.463 |
| DeBERTa | **0.660** | 0.606 | 0.698 | 0.656 | 0.347 | **0.528** | 0.566 |
| TAW | 0.609 | **0.642** | 0.664 | 0.644 | 0.360 | 0.428 | 0.558 |
| TAG | 0.623 | 0.618 | **0.727** | 0.647 | **0.423** | 0.479 | 0.586 |
| TATA | 0.638 | 0.629 | 0.654 | **0.669** | 0.416 | 0.521 | **0.588** |

Table 6: Macro average of the *Pro* and *Against* classes for the SEM16t6 dataset for our set of trained models. We bold the highest/best score in each column.

sults, we evaluate our model on the Zero-shot SEM16t6 dataset (Mohammad et al., 2016) which contains passages on six pre-defined topics: Donald Trump (DT), Hillary Clinton (HC), Feminist Movement (FM), Legalization of Abortion (LA), Atheism (A), and Climate Change (CC). Each passage's stance can again be classified in $\{Pro, Neutral, Against\}$. We give dataset statistics for the SEM16t6 dataset in Table 5.

We note that given that our TAG embeddings were trained on the VAST dataset that contained SEM16t6 topics, we retrain our TAG embedding layer on a reduced dataset that removed these SEM16t6 topics (30 training instances removed). Further, given that our TAG and TATA models werere implicitly trained on the rest of the VAST dataset (through the TAG embedding layer), in order to give a fairer comparison, we combine the reduced VAST (*i.e.*, without the SEM16t6 topics) and the SEM16t6 training sets when training each model. For each topic, we use the leave-one-target-out evaluation setup (Allaway and McKeown, 2023; Liang et al., 2022b) and report the macro average of the *Pro* and *Against* classes. As seen in Table 6, our TATA model, on average, again outperforms our other models and baseline models.

# 8   Conclusion

Combining pre-trained layers that learn topic-agnostic/TAG features and topic-aware/TAW features, our TATA model achieves state-of-the-art performance on the VAST test dataset. TATA performs better than prior works not only in Few-shot settings but also in Zero-shot settings. Our most common failure mode is cases where a given passage has both positive and negative stances within it. For future work, during training, we plan to incorporate additional passages with multiple stances toward different topics.

## Limitations

We note several limitations of our approach here:
**Use of News Articles for TAW Layer.** We note that while our use of news articles from a variety of sources enables us to build an informative latent across a variety of topics, our dataset is probably largely biased against topics that are not regularly spoken about in the news such as different science concepts or medical information.
**Use of Paraphrases for TAG and Noisy Topic Identification.** We note that using paraphrases while developing our augmented VAST dataset may unintentionally change the meaning of the original text. While we guard against this by utilizing conservative parameters of our `Dipper Paraphraser`, we acknowledge that it may result in incorrect information in our augmented VAST dataset.
**Zero-Shot vs. Few-Shot Setting Performance.** We note that, as in prior work such as CKE-Net (Liu et al., 2021a) and JointCL (Liang et al., 2022b), we achieve better results in a Zero-shot setting than in a Few-shot setting. This may indicate partially disagreeing labels among multiple annotators present within the original training VAST dataset that may be biasing the results for the topics in the Few-shot setting.

## Ethics Statement

We utilize a dataset of public news articles collected by Hanley and Durumeric (2023) to train a topic-aware encoding layer. We ask for permission to utilize and release sections of their dataset.

We note that our work can be utilized to understand the stance of news articles and comments on various phenomena ranging from individual political figures like Joe Biden or Donald Trump to topics such as immigration and vaccines. However, while our work substantially improves upon past works for stance detection, before it can be utilized or deployed, at scale, the accuracy for *Pro* and *Against* classes still needs to be improved as we still only achieve near $0.70$ $F_1$ scores for those classes.

## Acknowledgements

This work was supported in part by a gift from Google, Inc., the NSF Graduate Fellowship DGE-1656518, a Stanford Graduate Fellowship, and the Meta Ph.D. Fellowship in Computational Social Science.

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

# A  T5-Flan

Rather than considering all noun phrases in the subject and object sections of passages as potential topics as in Allaway and McKeown (2020), we instead prompt T5-Flan-Xl (Chung et al., 2022), an 11 billion parameter fine-tuned large language model to identify the topic for a given article passage given a list of candidate noun phrases extracted by `Spacy`. The T5-Flan-XL model that we utilize was fine-tuned on 193 tasks present in the T0 dataset (Sanh et al., 2022), which includes topic classification. Given how T5-Flan-XL was fine-tuned, we utilize the following prompt to extract the topic using the noun-phrases from the text as potential options: `Select the topic that this about:{text}\n\n{options}\n\n,{answer}`.

# B  Parrot Paraphraser

We utilize the `Parrot Paraphraser` (Damodaran, 2021) to generate paraphrases of topics. We utilize the `Parrot Paraphraser` for our topic paraphrases as it was fine-tuned on several shorter datasets. The `Parrot Paraprhaser`[5] was trained to develop semantic similar and logically consistent paraphrases by fine-tuning on NLU tasks. The `Parrot` paraphrase has input parameters for:

---

[5] https://huggingface.co/prithivida/parrot_paraphraser_on_T5

- Adequacy: Is the output semantically equivalent to the input?

- Fluency: How fluent is the English of the paraphrase?

- Diversity: How much has paraphrasing changed the original input?

We use the default setting for the `Parrot` Paraphrser, only changing the fluency to a value of 0.50 given that we input topic phrases rather than full sentences.

## C  Dipper Paraphraser

We utilize the `Dipper` (Krishna et al., 2023) Paraphraser[6], a T5-based model (Raffel et al., 2020) in order to paraphrase the original text in the `VAST` dataset. The `Dipper` paraphrase model was trained to give paraphrases of paragraph-level data by utilizing different translations of non-English novels into English aligned at a paragraph level. The `Dipper` model in addition has two parameters $L$ and $O$ that correspond to the estimated lexical diversity (using unigram token overlap) and the ordering of words (using the Kendall-Tau correlation of tokens). This is such that a $L = 40$ value corresponds to a 40% lexical modification of the original text and similarly for $O$.

The authors of `Dipper` found that using a high lexical diversity value (on a scale of 0 to 100) in the Dipper model has a tendency to change the proper nouns within a given paragraph. Given that for our pretraining of the TAG embedding layer, we require the stance of the original text to be maintained toward the nouns within it, as recommended by Krishna et al. (2023), we use a low lexical diversity value of 20. We use an order diversity of $O = 40$.

---

[6]https://huggingface.co/kalpeshk2011/dipper-paraphraser-xxl

# D    Example Parrot Paraphrases

| Original Topic | Paraprhase List |
| --- | --- |
| israelis' retributions | the retribution of Israel , israel's revenge |
| farm aide | aide, a farm assistant, farm support |
| taxing the rich | taxation of the wealthy , taxing wealthy |
| favors for one another | favors for each other, a favor for each other |
| government funding | source of money, government grants,federal funding, government money |
| the royals | the royal family, royalty |
| pain pills | pain reliever, pain medication, pain relief pills |
| illness | medical condition, disease, ill |
| china's manufacturing | manufacturing in china, china's production |
| people being obese | obesity, obese, people who are fat, people who are overweight |

Table 7: Examples of different topic paraphrases generated by the `Parrot` paraphraser (Damodaran, 2021).

# E   Example Dipper Paraphrases

| Original Text | Paraprhase |
|---|---|
| Absolutely it's needs to be defined and regulated in its use, as currently the word 'natural' when used on food products is totally confusing and meaningless. Clearly they are trying to imply the item is 'healthy' or possibly 'organic', but when you see food 'manufacturers' like Frito-Lay or Campbell's with products labelled 'natural', that alone should set off alarms that all is not what it seems. ;-) | The word "natural" has absolutely to be defined and regulated in its use. Applied to food products, the word is now so confusing and meaningless that it's useless. The mere fact that a food "manufacturer" like Frito-Lay or a can manufacturer like the Campbell Soup Company label their products as "natural" should alert you to the fact that not all is as it seems. Obviously they are trying to suggest that the product is "healthy," or possibly "organic." —Suomi-Maestro! |
| ..one must ask how much money they must make to demonstrate that they are among the best managed companies on the planet." They must make enough money to insure that they can never fail and threaten the stability of the worlds economy again. That much money. | ... to see that they are one of the best-managed companies on the planet, one must ask how much profit they must make.' If you wish to keep the economy stable you must never again threaten to go bankrupt. So you must make enough money to guarantee that. This much money. |
| I had a beautiful home birth with a licensed midwife in 2011. I was so inspired by the experience that I became a certified birth doula and have attended 17 births (many at home or at a birth center). All women should have the opportunity to choose where they feel safest birthing. SO grateful for midwifery! | In 2011, I had a beautiful homebirth with a licensed midwife. I was so enthusiastic about the whole thing that I became a certified birth doula and have been to seventeen more births since then, mostly at home or in a birthing center. "The women should be able to choose where they want to give birth. The gratefulness for midwifery! |
| "As a single mother I adopted my nearly two year old daughter from China at 49 and three years later I brought home my son from China – and he was nearly 13. Today he is one semester away from graduating college and she just had her Bat Mitzvah and is a rising eighth grader. Okay, so I am the oldest mom in her class – in her school. Its not my fault. These are the children I was meant to have and I just had to wait for them to be available so I could bring them home. One of the best things about being an older parent – one really doesn't care what others think. | In my forties I adopted my daughter from China. She was nearly two when she came to live with us. Three years later I brought home my son, then aged thirteen. Then she had her Bat Mitzvah and began the eighth grade. He was only one semester away from graduating from college. I am the oldest mother in her class—in her school. It is not my fault. 'No, I am not sorry,' he cried, 'I only wish that I could have my own children. I have been waiting for so long for them to become free so that I could take them home.' As an old parent one is no longer concerned with the opinions of others. |
| In considering the threat posed by Iran, it's important to remember that Iran has not invaded any of its neighbors. We have supported Iraq's invasion of Iran, under Saddam and even assisted Iraq in targeting chemical weapons attacks upon Iranian forces. We also supported the Shah, throughout his tenure, including over-throwing an elected government and installing the Shah. Iran has far more reason to distrust and fear us than the other way around. It also needs to be understood that Iran is presently in Iraq and Syria at the invitation of the official governments of those nations (with the Iraqi leader having basically been chosen by us), while the Syrian government does not want us in Syria and the Iraqi government is ambivalent at best about our presence. Waging combat through surrogates is actually the approach of a weak and cautious nation, not a strong and confident regional power. In terms of Americans and Westerners killed and threatened, Saudi Arabia is a far worse actor than Iran. In terms of the killing in Syria, Saudi Arabia is also a far worse actor than Iran. | In considering the threat posed by Iran it is important to remember that Iran has never invaded any of its neighbours. Iraq, under Saddam, invaded Iran and we even helped Iraq to attack Iran with chemical weapons. Throughout the Shah's reign, we supported him. We also supported his overthrowing an elected government and putting himself in power. In fact, we have far more reason to distrust and fear Iran than the other way around. In fact, Iran is in Syria and Iraq now at the official invitation of their respective governments (the one in Baghdad having basically been chosen by us), while the Syrian government does not want us in Syria and the Iraqi government at best is ambivalent about our presence there. A weak and wary nation is in fact the true adversary of a strong and confident local power. But the kingdom of Saudi Arabia is worse than Iran in killing and threatening Americans and Westerners. Certainly in the slaughter of Syria, Saudi Arabia is a far worse offender than Iran. |
| This is one of the most intelligent articles I have read on Syria. Hoffman is right – this is a civil war, and only the Syrians can resolve it. We need to stay out of this - it is not our concern. | The most intelligent article on Syria I have ever read. ... But it's a civil war. Only the Syrians can solve it. 'This isn't our business. We need to stay out of this. |
| Do ANY programs make money after taking into account: Amortization of physical plant including all athletic venues and buildings occupied exclusively by coaches, staff and athletes. Maintenance, heating, cooling, groundskeeping, police & fire protection for all athletic facilities and the territory that surrounds them. Administrator time spent dealing with athletic scandals, criminal athletes, and spoiled coaches. Subsidized travel to post season games by administrators and board members. Also consider: The total amount of tax deductible contributions to higher ed is likely to be a relatively fixed sum within any given set of tax laws. Therefore athletic department contributions subtract from contributions to legitimate academic purposes and should be counted as a cost, not a benefit. | Among the costs of a university program, you should include the depreciation of the physical plant, that is, all the sports arenas and the buildings occupied exclusively by coaches, staff and athletes. Do any of these programs show a profit, after you have included the depreciation? For all sports venues and their surroundings, maintenance, heating, air-conditioning, gardening, police, and fire protection. At the same time, the time of the manager is spent on athletic scandals, criminal athletes, and spoiled coaches. Administrators and board members traveling to post-season games. Contributions to higher education are likely to be relatively fixed within any given set of tax laws. Consider also the following. Contributions to the athletic department detract from contributions to legitimate academic purposes and should therefore be regarded as a cost, not a benefit. |

Table 8: Examples of different paraphrases of texts from the VAST dataset generated by the Dipper paraphraser (Krishna et al., 2023).