# OpenReview forum: "TATA: Stance Detection via Topic-Agnostic and Topic-Aware Embeddings"
_EMNLP/2023/Conference — EMNLP 2023 Main_

### Official Review · Reviewer_dEgC · 2023-08-03

**Typos Grammar Style And Presentation Improvements:** 1. This article has a lot of grammar …
**Soundness:** 2

**Excitement:**

2: Mediocre: This paper makes marginal contributions (vs non-contemporaneous work), so I would rather not see it in the conference.

**Missing References:**

None

**Paper Topic And Main Contributions:**

This paper mainly pre-trains two embedding layers to extract topic-agnostic and topic-aware features and utilizes these two kinds of features for zero-shot stance detection. Specifically, it constructs a new dataset including news articles and topics for pre-training the topic-aware embedding layer, and expands the original VAST dataset with paraphrases for pre-training the topic-agnostic embedding layer.

**Questions For The Authors:**

1. Could you further explain the design of attention calculation between h_taw and h_topic, as well as the meaning of r_topic?

**Reasons To Accept:**

1. This paper constructs a new dataset and expands the existing dataset for pre-training.
2. The proposed method achieves better performance on zero-shot stance detection.

**Reasons To Reject:**

1. The idea of topic-agnostic and topic-aware features itself has been proposed in previous work, thus this work is incremental to further advance this idea.
2. The main contributions of this work are still vague, which makes it hard to assess how solid this work is in advancing the research area.
3. The overall presentation of the paper is problematic with many typos and mistakes.

**Reproducibility:**

3: Could reproduce the results with some difficulty. The settings of parameters are underspecified or subjectively determined; the training/evaluation data are not widely available.

**Reviewer Confidence:**

4: Quite sure. I tried to check the important points carefully. It's unlikely, though conceivable, that I missed something that should affect my ratings.

---

> ### Author Rebuttal · Authors · 2023-08-28
>
> # Reviewer dEgC
> We first would like to thank Reviewer dEgC for their valuable feedback. For our rebuttal, we begin by describing why we decided to use two separate pre-trained layers for calculating topic-agnostic and topic-aware embeddings. We then describe in further detail the design of the attention mechanisms behind our TAW model.  We further describe several additional steps that we took to bolster our work's reproducibility as well as some new experiments that we performed.
>
> ## Our Use of Two Separate Layers
> We take a topic-agnostic and topic-aware approach given that we wish to take advantage of the two separate perspectives of determining stance (i.e. looking at generalized stance features that are common across different topics *and* trying to derive specific stance features particular to given (both seen and unseen) topics based on learned topic-related information). We now include this reasoning in our introduction and at the beginning of section 5 (methodology section). The reason why we take to developing separate embeddings for TAG and TAW is that the main crux of our approach is developing a topic latent that takes into account tens of thousands of diverse topics. As such, we cannot train these two together because we do not have stance labels for the passage attached to the topics we gather. As such we train these separately. We further note that while our TAW layer enables us to gather specific representations of each topic/target, our TAG layer in contrast enables us to design generalized stance features that are common within each stance class. By combining the generalized features and the rich topic embeddings, and training a network to utilize a combination of these perspectives, as shown by our results, we manage to further improve the generalization of our approach to unseen topics compared to past work.
>
> ## Design of Topic Aware/TAW Layer
>
> We note that prior works such as Alloway et al.[1] and JointCl [3] have focused on utilizing generalized topic representations by clustering the VAST dataset. These latent topic representations that they and we design, contain information about the topic relationships without requiring explicit human knowledge, allowing our approaches to extend to topics unseen in our VAST stance dataset. However, in previous approaches, by clustering the VAST dataset, these generalized representations were limited to the topics present within the dataset (i.e. those for which there were stance labels). However, by instead using an unstructured dataset and by first building a latent space where similar topics have similar representations and different ones have different representations, we manage to greatly expand the number of topics that we train on beyond the 4,641 present in the VAST training dataset to over 100,000. We further note that given our use of unstructured and unlabeled data, our approach is not limited to only these 100,000 topics. We now include this reasoning in our introduction and at the beginning of section 5 (methodology section).
>
> We note that while training our TAW layer and while performing our contrastive learning objective to create our topic latent, we utilized as positive pairs, items that had different texts and two different paraphrased topics (e.g. 1: [CLS] Drinking cups are the great [SEP] Cups [SEP] and 2: [CLS] Drinking containers are awesome [SEP] Mugs [SEP]. We utilized different paraphrased topics, to ensure that our TAW layer did not just learn to just move together items with the same exact same topic/target (e.g. only "cup" and "cup" vs. "cup" and "mug").  Rather, as previously noted, we wished for our topic latent to learn similar representations to items with similar and rephrased topics. However, given that we utilize different but similar topics in our contrastive learning step, we wish to ensure that the similar topics only have similar embeddings rather than one single representative cluster embedding. Triplet loss, which is used for contrastive learning and has been shown to produce informative latent spaces [4], allows for this intra-class variance. This is because Triplet loss only ensures that negative examples are further away in the latent space than positive examples. As such we utilized a Triplet loss within this training step. Once trained, this topic latent, which implicitly contains the relationship between topics, can then be utilized to get informative generalized topic representations for use in training a stance classifier, which heavily depends on the features of the topic/target.
>
>
>
> ## Design of Attention in TAW network
> We note that the place of h\_taw and h\_topic were switched within our original document. This has been fixed. The purpose of the attention layer within our model is to determine the relationship between the DeBERTa representation of the topic/target t and the latent representation of the topic. In this way, we manage to better determine which aspect of the latent space representation of the topic is important in representing a particular instance of a topic/target. We thus represent this as the representation r\_topic that captures the relationship between the topic/target t and its generalized latent space representation h\_taw. We utilize a scaled dot product attention as proposed in Vaswani et al. [3].
>
> ## Stylistic Improvements
> We have fixed the errors outlined by Reviewer dEgC. We have already taken several passes at our paper to ensure that we clean up the errors and stylistic problems. We further utilized Grammarly to ensure all spelling mistakes were fixed. In addition, we took the following steps to polish and fix errors within the original document:
>
> * The use of "passage" versus "text" when referring to the documents within the VAST dataset has been standardized to only use "passage".
> * All mentions of VAST dataset now have the same stylized text (using texttt)
> * We note that the place of h\_taw and h\_topic were switched within our original document on lines 414-415. This has been fixed.
> * We redid Figure 1 so that all errors have been fixed. We have removed the Microsoft images (Microsoft developed DeBERTa, hence our prior use of them), opting for a simpler and cleaner trapezoid encoder image.
>
>
> ## Improved Reproducibility
>
> We note that all data and models that we utilized for this work are open-sourced. Since the submission of this paper, Hanley et al. [5] have created a means (by emailing the authors) on their GitHub for requesting their dataset. All other datasets that we utilized within this work (VAST, SemEval16) are public or were generated using the public Parrot and Dipper models. However, we plan to release the paraphrase datasets we utilized to train our TAG and TAW upon publication (although similar datasets can be generated using the Parrot and Dipper huggingface models). Finally, to improve the consistency and robustness of our results, we now train each model with a different random seed a total of five times and report the average of those five different runs. We present these updated results below:
>
> |Model|Zero-Shot VAST |Few-Shot VAST |
> |-----------|-----------|-----------|
> | |Pro, Con, Neutral, All|Pro, Con, Neutral, All|
> |-----------|-----------|-----------|
> |DeBERTa|0.680, 0.683,0.900,0.755 | 0.659,0.657,0.869,0.728|
> |-----------|-----------|-----------|
> |TAG|0.681, 0.687,0.901,0.756 | 0.665,0.655,0.868,0.729|
> |-----------|-----------|-----------|
> |TAW|0.672, 0.709,0.903,0.760 | 0.656,0.677,0.869,0.736|
> |-----------|-----------|-----------|
> |TATA|0.695,0.711,0.905,0.771|0.665,0.683,0.873,0.741|
> |-----------|-----------|-----------|
>
> |Model|Qte |Sarc|Imp|mlS|mlT|
> |-----------|-----------|-----------|-----------|-----------|-----------|
> |DeBERTa|0.709|0.713|0.667|	0.578|0.676|
> |-----------|-----------|-----------|-----------|-----------|-----------|
> |TAG|0.710|0.731|0.668|	0.568|0.676|
> |-----------|-----------|-----------|-----------|-----------|-----------|
> |TAW|0.712|0.722|0.683|	0.585|0.693|
> |-----------|-----------|-----------|-----------|-----------|-----------|
> |TATA|0.714|0.730|0.693|	0.603|0.704|
> |-----------|-----------|-----------|-----------|-----------|-----------|
>
>
> We again observe that all of our models perform better than the previous state-of-the-art JointCl (0.723 in Zero-shot and 0.715 in the Few-shot) [2]. Furthermore, our TAG and TAW models perform better than our DeBERTa model in the Zero-shot setting evidencing that our approach *does* help for stance detection on unseen topics. As seen, we again see that our TATA model performs the best in both the Zero-shot and the Few-shot settings.
>
> ## Additional Experiments
>
>
> ### Few-shot Bias
> We note that for our results, our models perform better in the Zero-shot setting. In order to ascertain whether the number of few-shot examples for particular topics in the VAST dataset affects the performance of our models during inference, we perform another experiment; namely we measure whether the number of lexically similar training examples with regards to a topic/target within VAST (e.g. where a lexically similar topic is as previously defined in our paper and Alloway et al. [2] as having a GloVe embedding cosine similarity above 0.9) is correlated with whether our system correctly labels a given instance of that target/topic. If there is a positive correlation this may indicate that the topics with few examples may be noisy/biasing our results (as those with no similar training examples at all have higher F1-scores than those in the Few-shot setting). In the few-shot setting, we find a slight Pearson correlation (0.186, p-value =0.021) between the number of semantic similar training examples and the percentage of a given topic's test examples that are labeled correctly by our TATA model (0.2675, p-value=0.0008 for TAG model; 0.199, p-value=0.012 for TAW model; 0.215, p-value= 0.007 for deBERTa model). This suggests some noisiness or a tendency of our model to be biased on topics in VAST that only have one or two training examples. As reported by Alloway et al.[3], there is a high, but not perfect agreement among labelers (75%), which may also help explain this phenomenon. We now include this discussion within our paper following the *Correlation with Inclusion in TAW dataset* section.
>
> ### Zero-shot Stance Detection on the SemEval dataset
> We now as suggested by Reviewer epNc, separately retrain our TAG layer without training instances that have SemEval16 topics (Hillary Clinton, Donald Trump, Legalization of Abortion, Climate Change, Atheism, Feminist Movement) in order to test our TAG and TATA models in a Zero-shot setting on the SemEval16 topics.  We note that these topics only consisted of 30 training instances (out of 13,477) in the original VAST dataset. We note that given that our TAG and TATA models are implicitly trained on the rest of the VAST dataset (through the TAG encoding layer), we now, in order to give a fairer comparison, combine the reduced VAST (i.e. without the SemEval16 topics) and the SemEval16 training sets when training each model. We give the results below.
>
> |Model|A|CC	|LA|FM|	HC|	DT|	Avg|
> |-----------|-----------|-----------|-----------|-----------|-----------|-----------|-----------|
> |DeBERTa|0.528|0.347|0.606|	0.656	|0.698	|0.660|	0.566|
> |-----------|-----------|-----------|-----------|-----------|-----------|-----------|-----------|
> |TAG|0.479|0.423|0.618|	0.647|0.727	|0.623|	0.586|
> |-----------|-----------|-----------|-----------|-----------|-----------|-----------|-----------|
> |TAW|0.428|0.360|0.642|0.644|0.664|0.609|0.558|
> |-----------|-----------|-----------|-----------|-----------|-----------|-----------|-----------|
> |TATA|0.521|0.416|0.629|0.669|0.654|0.638|0.588|
> |-----------|-----------|-----------|-----------|-----------|-----------|-----------|-----------|
>
>
> We again observe that our TATA model performs better within this setting compared to our DeBERTa model ( as well as previous state-of-the-art models which achieved an average score near 0.50 [e.g. JointCL] [2]).
>
>
> ### Topic Aware/TAW Layer Potentially Just Utilizing the Topic/Target
> As suggested by Reviewer epNc, we perform an additional ablation study to ensure that our TAW subnetwork does not simply utilize the topics to predict the stance of each example. As such we ran a study where for the Few and Zero-shot test sets within the the VAST dataset, we predicted the stance based on the topic. To do so, for our TAW encoding layer rather than passing [CLS] p\_i [SEP] t\_i [SEP] to the the layer as input, we pass in [CLS]  [SEP] t\_i [SEP].
>
> |Model|Zero-Shot VAST |Few-Shot VAST |
> |-----------|-----------|-----------|
> | |Pro, Con, Neutral, All|Pro, Con, Neutral, All|
> |-----------|-----------|-----------|
> |TAW|0.034,0.000,0.519,0.184 | 0.020,0.000,0.507,0.176
> |-----------|-----------|-----------|
>
> As seen in the above table, just predicting based on the topic significantly hurts our results in the few-shot and zero-shot settings, with our model nearly always predicting the Neutral class. This behavior largely conforms to the definition of the Neutral class; without any input text, the stance of an empty string to a given topic should be neutral.  This suggests that our TAW model is not only picking up on the topic but is also heavily considering the text when predicting the stance.
>
> ## References
>
> [1] Emily Allaway and Kathleen McKeown. 2020. Zero-shot stance detection: a dataset and model using
> generalized topic representations. arXiv preprint arXiv:2010.03640.
>
> [2] Bin Liang, Qinlin Zhu, Xiang Li, Min Yang, Lin Gui, Yulan He, and Ruifeng Xu. 2022b. Jointcl: A joint contrastive learning framework for zero-shot stance detection. In Proceedings of the 60th Annual Meeting of the Association for Computational Linguistics (Volume 1: Long Papers), volume 1, pages 81–91. Association for Computational Linguistics.
>
> [3] Ashish Vaswani, Noam Shazeer, Niki Parmar, Jakob Uszkoreit, Llion Jones, Aidan N Gomez, Łukasz Kaiser, and Illia Polosukhin. 2017. Attention is all you need. Advances in neural information processing systems, 30.
>
> [4] Hong Xuan, Abby Stylianou, and Robert Pless. 2020. Improved embeddings with easy positive triplet mining. In Proceedings of the IEEE/CVF Winter Conference on Applications of Computer Vision, pages
> 903 2474–2482.
>
>
> [5] Hans WA Hanley and Zakir Durumeric. 2023. Machine-made media: Monitoring the mobilization of machine-generated articles on misinformation and mainstream news websites. arXiv preprint arXiv:2305.09820.

---

### Official Review · Reviewer_TUpb · 2023-08-04

**Soundness:** 3

**Excitement:**

3: Ambivalent: It has merits (e.g., it reports state-of-the-art results, the idea is nice), but there are key weaknesses (e.g., it describes incremental work), and it can significantly benefit from another round of revision. However, I won't object to accepting it if my co-reviewers champion it.

**Paper Topic And Main Contributions:**

The paper proposes a novel stance detection mechanism called TATA (Topic-Agnostic and Topic-Aware) that addresses the challenge of generalizing stance detection to unseen topics. The authors combine contrastive learning with an unlabeled dataset of news articles to train topic-agnostic (TAG) and topic-aware (TAW) embeddings. The embeddings are then utilized in a full TATA model, achieving state-of-the-art performance on several public stance-detection datasets. The paper presents a clear methodology and achieves promising results.

**Reasons To Accept:**

1. The paper tackles an important problem in stance detection and proposes an effective solution.
2. The use of contrastive learning and the incorporation of topic-aware features demonstrate the authors' innovation in addressing the challenges of generalization to unseen topics.
3. The TATA model achieves state-of-the-art results on zero-shot and few-shot stance detection datasets, showing its effectiveness.


**Reasons To Reject:**

1. Novelty of Topic Agnostic/TAG Encoding Layer and Topic-Aware/TAW Encoding Layer:

While the paper introduces the Topic Agnostic (TAG) and Topic-Aware (TAW) encoding layers as novel components of the TATA model, there are a few aspects that could be further clarified or expanded upon to strengthen their novelty:

a. The rationale behind choosing contrastive learning for the TAG encoding layer needs more in-depth explanation. The paper briefly mentions that contrastive learning allows similar embeddings for instances with the same stance and different embeddings for instances with different stances. However, more discussion on how this approach improves stance feature modeling compared to other methods would enhance the novelty explanation.

b. The use of a triplet loss in the TAW encoding layer for topic representation learning is intriguing. However, the paper could elaborate on why this approach is suitable for handling the noisy TAW dataset and how it effectively generates rich latent representations of topics. A comparison with alternative methods for topic-awareness in stance detection would provide better insight into the novelty of this approach.

2. Analysis of Experiments:

While the paper reports state-of-the-art performance on zero-shot and few-shot stance detection datasets, the analysis of the experiments could be more comprehensive。


**Reproducibility:**

3: Could reproduce the results with some difficulty. The settings of parameters are underspecified or subjectively determined; the training/evaluation data are not widely available.

**Reviewer Confidence:**

3: Pretty sure, but there's a chance I missed something. Although I have a good feel for this area in general, I did not carefully check the paper's details, e.g., the math, experimental design, or novelty.

---

> ### Author Rebuttal · Authors · 2023-08-28
>
> # Reviewer TUpb
> We first would like to thank Reviewer TUpb for their valuable feedback. For our rebuttal, we begin by describing several details for why we chose contrastive learning and triplet losses for our topic-agnostic and topic-aware embedding layers. We further describe several additional steps that we took to bolster our work's reproducibility as well as some new experiments that we performed.
>
>
> ## Choice of Contrastive Learning for Topic-Agnostic/TAG Layer
> Zero-shot stance detection seeks to identify the stance of texts toward targets/topics that were not seen during training [1]. One intuitive way of identifying the stance of a given text toward a topic/target is to train a model that identifies relevant generalized per-class stance features from text-target pairs that can then be utilized during inference when the target/topic was not seen during training. As found in prior works [2], contrastive learning can be utilized to create robust feature representations within classification tasks. Given our need to apply these features to topics/targets outside of our training and the use of contrastive learning for creating hard distinctions between different classes, we thus apply this methodology for creating generalized stance features which may be useful for predicting the stance of unseen topics. We thus *do* train a network where each stance class is placed in a distinct embedding space regardless of its topic so that features can be used later to train a  model that works on unseen topics. We now include this reasoning in our introduction and at the beginning of section 5 (methodology section).
>
> ## Choice of Triplet Loss for Topic Aware/TAW Layer
> We note that prior works such as Alloway et al. [1] and JointCl [3] have focused on utilizing generalized topic representations by clustering the VAST dataset. These latent topic representations that they and we design, contain information about the topic relationships without requiring explicit human knowledge, allowing our approaches to extend to topics unseen in our VAST stance dataset. However, in previous approaches, by clustering the VAST dataset, these generalized representations were limited to the topics present within the dataset (i.e. those for which there were stance labels). However, by instead using an unstructured dataset and by first building a latent space where similar topics have similar representations and different ones have different representations, we manage to greatly expand the number of topics that we train on beyond the 4,641 present in the VAST training dataset to over 100,000. We further note that given our use of unstructured and unlabeled data, our approach is not limited to only these 100,000 topics. We now include this reasoning in our introduction and at the beginning of section 5 (methodology section).
>
> We note that while training our TAW layer and while performing our contrastive learning objective to create our topic latent, we utilized as positive pairs, items that had different texts and two different paraphrased topics (e.g. 1: [CLS] Drinking cups are the great [SEP] Cups [SEP] and 2: [CLS] Drinking containers are awesome [SEP] Mugs [SEP]. We utilized different paraphrased topics, to ensure that our TAW layer did not just learn to just move together items with the same exact same topic/target (e.g. only "cup" and "cup" vs. "cup" and "mug").  Rather, as previously noted, we wished for our topic latent to learn similar representations to items with similar and rephrased topics. However, given that we utilize different but similar topics in our contrastive learning step, we wish to ensure that the similar topics only have similar embeddings rather than one single representative cluster embedding. Triplet loss, which is used for contrastive learning and has been shown to produce informative latent spaces [4], allows for this intra-class variance. This is because Triplet loss only ensures that negative examples are further away in the latent space than positive examples. As such we utilized a Triplet loss within this training step. Once trained, this topic latent, which implicitly contains the relationship between topics, can then be utilized to get informative generalized topic representations for use in training a stance classifier, which heavily depends on the features of the topic/target.
>
> ## Stylistic Improvements
> We have fixed the errors outlined by the other reviewers. We have already taken several passes at our paper to ensure that we clean up the errors and stylistic problems. We further utilized Grammarly to ensure all spelling mistakes were fixed. In addition, we took the following steps to polish and fix errors within the original document:
>
> * The use of "passage" versus "text" when referring to the documents within the VAST dataset has been standardized to only use "passage".
> * All mentions of VAST dataset now have the same stylized text (using texttt)
> * We note that the place of h\_taw and h\_topic were switched within our original document on lines 414-415. This has been fixed.
> * We redid Figure 1 so that all errors have been fixed. We have removed the Microsoft images (Microsoft developed DeBERTa, hence our prior use of them), opting for a simpler and cleaner trapezoid encoder image.
>
> ## Improved Reproducibility
>
> We note that all data and models that we utilized for this work are open-sourced. Since the submission of this paper, Hanley et al. [5] have created a means (by emailing the authors) on their paper's GitHub for requesting their dataset. All other datasets that we utilized within this work (VAST, SemEval16) are public or were generated using the public Parrot and Dipper models. However, we plan to release the paraphrase datasets we utilized to train our TAG and TAW upon publication (although similar datasets can be generated using the Parrot and Dipper huggingface models). Finally, to improve the consistency and robustness of our results, we now train each model with a different random seed a total of five times and report the average of those five different runs. We present these updated results below:
>
> |Model|Zero-Shot VAST |Few-Shot VAST |
> |-----------|-----------|-----------|
> | |Pro, Con, Neutral, All|Pro, Con, Neutral, All|
> |-----------|-----------|-----------|
> |DeBERTa|0.680, 0.683,0.900,0.755 | 0.659,0.657,0.869,0.728|
> |-----------|-----------|-----------|
> |TAG|0.681, 0.687,0.901,0.756 | 0.665,0.655,0.868,0.729|
> |-----------|-----------|-----------|
> |TAW|0.672, 0.709,0.903,0.760 | 0.656,0.677,0.869,0.736|
> |-----------|-----------|-----------|
> |TATA|0.695,0.711,0.905,0.771|0.665,0.683,0.873,0.741|
> |-----------|-----------|-----------|
>
> |Model|Qte |Sarc|Imp|mlS|mlT|
> |-----------|-----------|-----------|-----------|-----------|-----------|
> |DeBERTa|0.709|0.713|0.667|	0.578|0.676|
> |-----------|-----------|-----------|-----------|-----------|-----------|
> |TAG|0.710|0.731|0.668|	0.568|0.676|
> |-----------|-----------|-----------|-----------|-----------|-----------|
> |TAW|0.712|0.722|0.683|	0.585|0.693|
> |-----------|-----------|-----------|-----------|-----------|-----------|
> |TATA|0.714|0.730|0.693|	0.603|0.704|
> |-----------|-----------|-----------|-----------|-----------|-----------|
>
> We again observe that all of our models perform better than the previous state-of-the-art JointCl (0.723 in Zero-shot and 0.715 in the Few-shot) [3]. Furthermore, our TAG and TAW models perform better than our DeBERTa model in the Zero-shot setting evidencing that our approach *does* help for stance detection on unseen topics. As seen, we again see that our TATA model performs the best in both the Zero-shot and the Few-shot settings.
>
> ## Additional Experiments
>
>
> ### Few-shot Bias
> We note that for our results, our models perform better in the Zero-shot setting. In order to ascertain whether the number of few-shot examples for particular topics in the VAST dataset affects the performance of our models during inference, we perform another experiment; namely we measure whether the number of lexically similar training examples with regards to a topic/target within VAST (e.g. where a lexically similar topic is as previously defined in our paper and Alloway et al. [2] as having a GloVe embedding cosine similarity above 0.9) is correlated with whether our system correctly labels a given instance of that target/topic. If there is a positive correlation this may indicate that the topics with few examples may be noisy/biasing our results (as those with no similar training examples at all have higher F1-scores than those in the Few-shot setting). In the few-shot setting, we find a slight Pearson correlation (0.186, p-value =0.021) between the number of semantic similar training examples and the percentage of a given topic's test examples that are labeled correctly by our TATA model (0.2675, p-value=0.0008 for TAG model; 0.199, p-value=0.012 for TAW model; 0.215, p-value= 0.007 for deBERTa model). This suggests some noisiness or a tendency of our model to be biased on topics in VAST that only have one or two training examples. As reported by Alloway et al.[3], there is a high, but not perfect agreement among labelers (75%), which may also help explain this phenomenon. We now include this discussion within our paper following the *Correlation with Inclusion in TAW dataset* section.
>
> ### Zero-shot Stance Detection on the SemEval dataset
> We now as suggested by Reviewer epNc, separately retrain our TAG layer without training instances that have SemEval16 topics (Hillary Clinton, Donald Trump, Legalization of Abortion, Climate Change, Atheism, Feminist Movement) in order to test our TAG and TATA models in a Zero-shot setting on the SemEval16 topics.  We note that these topics only consisted of 30 training instances (out of 13,477) in the original VAST dataset. We note that given that our TAG and TATA models are implicitly trained on the rest of the VAST dataset (through the TAG encoding layer), we now, in order to give a fairer comparison, combine the reduced VAST (i.e. without the SemEval16 topics) and the SemEval16 training sets when training each model. We give the results below.
>
> |Model|A|CC	|LA|FM|	HC|	DT|	Avg|
> |-----------|-----------|-----------|-----------|-----------|-----------|-----------|-----------|
> |DeBERTa|0.528|0.347|0.606|	0.656	|0.698	|0.660|	0.566|
> |-----------|-----------|-----------|-----------|-----------|-----------|-----------|-----------|
> |TAG|0.479|0.423|0.618|	0.647|0.727	|0.623|	0.586|
> |-----------|-----------|-----------|-----------|-----------|-----------|-----------|-----------|
> |TAW|0.428|0.360|0.642|0.644|0.664|0.609|0.558|
> |-----------|-----------|-----------|-----------|-----------|-----------|-----------|-----------|
> |TATA|0.521|0.416|0.629|0.669|0.654|0.638|0.588|
> |-----------|-----------|-----------|-----------|-----------|-----------|-----------|-----------|
>
>
> We again observe that our TATA model performs better within this setting compared to our DeBERTa model ( as well as previous state-of-the-art models which achieved an average score near 0.50 [e.g. JointCL] [3]).
>
>
> ### Topic Aware/TAW Layer Potentially Just Utilizing the Topic/Target
> As suggested by Reviewer epNc, we perform an additional ablation study to ensure that our TAW subnetwork does not simply utilize the topics to predict the stance of each example. As such we ran a study where for the Few and Zero-shot test sets within the the VAST dataset, we predicted the stance based on the topic. To do so, for our TAW encoding layer rather than passing [CLS] p\_i [SEP] t\_i [SEP] to the the layer as input, we pass in [CLS]  [SEP] t\_i [SEP].
>
> |Model|Zero-Shot VAST |Few-Shot VAST |
> |-----------|-----------|-----------|
> | |Pro, Con, Neutral, All|Pro, Con, Neutral, All|
> |-----------|-----------|-----------|
> |TAW|0.034,0.000,0.519,0.184 | 0.020,0.000,0.507,0.176
> |-----------|-----------|-----------|
>
> As seen in the above table, just predicting based on the topic significantly hurts our results in the few-shot and zero-shot settings, with our model nearly always predicting the Neutral class. This behavior largely conforms to the definition of the Neutral class; without any input text, the stance of an empty string to a given topic should be neutral.  This suggests that our TAW model is not only picking up on the topic but is also heavily considering the text when predicting the stance.
>
> ## References
>
> [1] Emily Allaway and Kathleen McKeown. 2020. Zero-shot stance detection: a dataset and model using
> generalized topic representations. arXiv preprint arXiv:2010.03640.
>
> [2] Nikunj Saunshi, Orestis Plevrakis, Sanjeev Arora, Mikhail Khodak, and Hrishikesh Khandeparkar.2019. A theoretical analysis of contrastive unsupervised representation learning. In International Conference on Machine Learning, pages 5628–5637. PMLR.
>
> [3] Bin Liang, Qinlin Zhu, Xiang Li, Min Yang, Lin Gui, Yulan He, and Ruifeng Xu. 2022b. Jointcl: A joint contrastive learning framework for zero-shot stance detection. In Proceedings of the 60th Annual Meeting of the Association for Computational Linguistics (Volume 1: Long Papers), volume 1, pages 81–91. Association for Computational Linguistics.
>
> [4] Hong Xuan, Abby Stylianou, and Robert Pless. 2020. Improved embeddings with easy positive triplet mining. In Proceedings of the IEEE/CVF Winter Conference on Applications of Computer Vision, pages
> 903 2474–2482.
>
> [5] Hans WA Hanley and Zakir Durumeric. 2023. Machine-made media: Monitoring the mobilization of machine-generated articles on misinformation and mainstream news websites. arXiv preprint arXiv:2305.09820.

---

### Official Review · Reviewer_epNc · 2023-08-05

**Soundness:** 4

**Excitement:**

3: Ambivalent: It has merits (e.g., it reports state-of-the-art results, the idea is nice), but there are key weaknesses (e.g., it describes incremental work), and it can significantly benefit from another round of revision. However, I won't object to accepting it if my co-reviewers champion it.

**Paper Topic And Main Contributions:**

This paper introduces a new architecture for stance detection based on building a contrastive network that learns topic-agnostic embeddings (irrespective of topics, two inputs that have the same class of stance favor/neutral/against share similar values in embedding space) and topic-aware embeddings (two inputs that belong to the same topic are placed closer in the embedding space, irrespective of stance). This in combination with an attention network can help achieve state-of-the-art performance of stance detection.


**Questions For The Authors:**

* A. For topic-agnostic setting, all of one stance seems to shares a similar embedding space– does it mean despite the topics?  Why is contrastive loss preferred in this case especially if topics are not the same?


**Reasons To Accept:**

* A combination of two models for stance detection such that one explicitly models the topic while the disregards the topic and models the target class is an interesting approach and can help improve stance detection to unseen, new topics.
* The organization is good and it is easy to understand the narrative of the paper.



**Reasons To Reject:**

* While being zeroshot better than fewshot is glossed away in the limitations section as being similar to other two works before this but in fact few shot is better than zero shot in most cases, and in cases where it is not, it is still better for some of classes- signifying an improvement. The macro average is almost always better for fewshot except for JointCL. The other models are able to improve with fewshot which means that the issue is not with the annotator disagreement, but maybe the model is probably overfit to the topic-agnostic model? While a more extensive comparison could be welcome, maybe it is good to include a discussion with results not including the topics from SemEval so that zeroshot could also be evaluated on SemEval as well, in the next version of the paper.
However, in this paper, fewshot is consistently worse by 0.02-0.05 which is a strange phenomenon and there needs to be a discussion on this more than just a point on Limitations.
* Topic-aware embeddings are trained with topics being one of the inputs (separated from text by a separator token), this could bias the TAW model to pick up on solely the topics, and topics could themselves be used to predict the stance. An attention map/ ablation of topic tokens/predicting based on topic alone from the input could be better to explore what the model focuses on.
* The paper needs to significantly be improved stylistically. There are many grammatical and spelling errors, along with messy figures which should definitely be worked on for the next version.


**Reproducibility:**

4: Could mostly reproduce the results, but there may be some variation because of sample variance or minor variations in their interpretation of the protocol or method.

**Reviewer Confidence:**

4: Quite sure. I tried to check the important points carefully. It's unlikely, though conceivable, that I missed something that should affect my ratings.

**Typos Grammar Style And Presentation Improvements:**

* Line 89: “Detection”
* Line 157: “topic-topic” just once?
* Line 159: t_i? Not t_i1?
* Line 246: “datasett”
* Line 276: comma after phrases (ambiguous otherwise)
* paraphrase is spelled as “paraprhase” in multiple locations
* Many other typos: spell-check recommended for the next version of the paper
* Fig 1 caption: v_{text|topic} – is it h instead? Inconsistent use of “text” and “passage”. The Microsoft logo colors are distracting and not necessary in the diagram, and it could be made cleaner and easier to understand.

---

> ### Author Rebuttal · Authors · 2023-08-28
>
> # Reviewer epNc
>
> We first would like to thank Reviewer epNc for their detailed feedback. For our rebuttal, we address the main issues raised by Reviewer epNc in turn, namely (1) the potential overfitting of our models on the topic-agnostic/TAG encodings, (2) implementing retraining of our models so that we may perform zero-shot stance detection on the SemEval16 dataset, (3) whether our TAW encoding layer potentially is only predicting based on the topic, (4) our choice of contrastive learning for the topic-agnostic/TAG encoding layer, and finally (5) the stylistic issues present within our original document. In addition to responding to these issues and describing several new experiments that we performed to address these issues, we further describe several additional steps that we took to bolster our work's reproducibility.
>
>
> ## Overfitting on Topic Agnostic Encodings: (Zero-shot vs. Few-shot)
> While we acknowledge that it is possible that our TATA model is overfitting the Topic Agnostic (TAG) embeddings, we note that this would largely not explain why our Topic-Aware (TAW)  subnetwork model also has a higher score within the Zero-shot setting. We further note, as reported in our paper and in the JointCL [1] paper,  CKE-Net (0.1 F1 worse), TGA-NET (0.2 F1 worse), JointCL (0.8 F1 worse), BERT (1.5 F1 worse), as well as our DeBERTa model (2.4 F1 worse), (which is fine-tuned with a classification head on top of DeBERTa model) all perform worse within the few-shot setting compared to the zero-shot setting [1].
>
> In order to ascertain whether the number of few-shot examples for particular topics in the VAST dataset affects the performance of our models during inference, we perform another experiment. In order to ascertain whether the number of few-shot examples for particular topics in the VAST dataset affects the performance of our models during inference, we perform another experiment; namely we measure whether the number of lexically similar training examples with regards to a topic/target within VAST (e.g. where a lexically similar topic is as previously defined in our paper and Alloway et al. [2] as having a GloVe embedding cosine similarity above 0.9) is correlated with whether our system correctly labels a given instance of that target/topic. If there is a positive correlation this may indicate that the topics with few examples may be noisy/biasing our results (as those with no similar training examples at all have higher F1-scores than those in the Few-shot setting). In the few-shot setting, we find a slight Pearson correlation (0.186, p-value =0.021) between the number of semantic similar training examples and the percentage of a given topic's test examples that are labeled correctly by our TATA model (0.2675, p-value=0.0008 for TAG model; 0.199, p-value=0.012 for TAW model; 0.215, p-value= 0.007 for deBERTa model). Together with the other models that perform worse in the Few-shot setting, this suggests some noisiness or a tendency of our model to be biased on topics in VAST that only have one or two training examples. As reported by Alloway et al.[3], there is a high, but not perfect agreement among labelers (75%), which may also help explain this phenomenon. We now include this discussion within our paper following the *Correlation with Inclusion in TAW dataset* section.
>
>
> ## Zero-shot Stance Detection on the SemEval16 dataset
> We now as suggested by Reviewer epNc, separately retrain our TAG layer without training instances that have SemEval16 topics (Hillary Clinton, Donald Trump, Legalization of Abortion, Climate Change, Atheism, Feminist Movement) in order to test our TAG and TATA models in a Zero-shot setting on the SemEval16 topics.  We note that these topics only consisted of 30 training instances (out of 13,477) in the original VAST dataset. We note that given that our TAG and TATA models are implicitly trained on the rest of the VAST dataset (through the TAG encoding layer), we now, in order to give a fairer comparison, combine the reduced VAST (i.e. without the SemEval16 topics) and the SemEval16 training sets when training each model. We give the results below.
>
> |Model|A|CC	|LA|FM|	HC|	DT|	Avg|
> |-----------|-----------|-----------|-----------|-----------|-----------|-----------|-----------|
> |DeBERTa|0.528|0.347|0.606|	0.656	|0.698	|0.660|	0.566|
> |-----------|-----------|-----------|-----------|-----------|-----------|-----------|-----------|
> |TAG|0.479|0.423|0.618|	0.647|0.727	|0.623|	0.586|
> |-----------|-----------|-----------|-----------|-----------|-----------|-----------|-----------|
> |TAW|0.428|0.360|0.642|0.644|0.664|0.609|0.558|
> |-----------|-----------|-----------|-----------|-----------|-----------|-----------|-----------|
> |TATA|0.521|0.416|0.629|0.669|0.654|0.638|0.588|
> |-----------|-----------|-----------|-----------|-----------|-----------|-----------|-----------|
>
>
> We again observe that our TATA model performs better within this setting compared to our DeBERTa model ( as well as previous state-of-the-art models which achieved an average score near 0.50 [e.g. JointCL] [1]).
>
>
> ## Topic Aware/TAW Layer Potentially Just Utilizing the Topic/Target
> As suggested by Reviewer epNc, we perform an additional ablation study to ensure that our TAW subnetwork does not simply utilize the topics to predict the stance of each example. As such we ran a study where for the Few and Zero-shot test sets within the the VAST dataset, we predicted the stance based on the topic. To do so, for our TAW encoding layer rather than passing [CLS] p\_i [SEP] t\_i [SEP] to the the layer as input, we pass in [CLS]  [SEP] t\_i [SEP].
>
> |Model|Zero-Shot VAST |Few-Shot VAST |
> |-----------|-----------|-----------|
> | |Pro, Con, Neutral, All|Pro, Con, Neutral, All|
> |-----------|-----------|-----------|
> |TAW|0.034,0.000,0.519,0.184 | 0.020,0.000,0.507,0.176
> |-----------|-----------|-----------|
>
> As seen in the above table, just predicting based on the topic significantly hurts our results in the few-shot and zero-shot settings, with our model nearly always predicting the Neutral class. This behavior largely conforms to the definition of the Neutral class; without any input text, the stance of an empty string to a given topic should be neutral.  This suggests that our TAW model is not only picking up on the topic but is also heavily considering the text when predicting the stance.
>
>
>
>
> ## Choice of Contrastive Learning for Topic-Agnostic/TAG Layer
> Zero-shot stance detection seeks to identify the stance of texts toward targets/topics that were not seen during training [2]. One intuitive way of identifying the stance of a given text toward a topic/target is to train a model that identifies relevant generalized per-class stance features from text-target pairs that can then be utilized during inference when the target/topic was not seen during training. As found in prior works [3], contrastive learning can be utilized to create robust feature representations within classification tasks. Given our need to apply these features to topics/targets outside of our training and the use of contrastive learning for creating hard distinctions between different classes, we thus apply this methodology for creating generalized stance features which may be useful for predicting the stance of unseen topics. We thus *do* train a network where each stance class is placed in a distinct embedding space regardless of its topic so that features can be used later to train a  model that works on unseen topics.
>
>
>
>
>
> ## Stylistic Improvements
> We have fixed the errors outlined by Reviewer epNc. We have already taken several passes at our paper to ensure that we clean up the errors and stylistic problems. We further utilized Grammarly to ensure all spelling mistakes were fixed. In addition, we took the following steps to polish and fix errors within the original document:
>
> * The use of "passage" versus "text" when referring to the documents within the VAST dataset has been standardized to only use "passage".
> * All mentions of VAST dataset now have the same stylized text (using texttt)
> * We note that the place of h\_taw and h\_topic were switched within our original document on lines 414-415. This has been fixed.
> * We redid Figure 1 so that all errors have been fixed. We have removed the Microsoft images (Microsoft developed DeBERTa, hence our prior use of them), opting for a simpler and cleaner trapezoid encoder image.
>
> ## Improved Reproducibility
>
> We note that all data and models that we utilized for this work are open-sourced. Since the submission of this paper, Hanley et al. [4] have created a means (by emailing the authors) on their paper's GitHub for requesting their dataset. All other datasets that we utilized within this work (VAST, SemEval16) are public or were generated using the public Parrot and Dipper models. However, we plan to release the paraphrase datasets we utilized to train our TAG and TAW upon publication (although similar datasets can be generated using the Parrot and Dipper huggingface models). Finally, to improve the consistency and robustness of our results, we now train each model with a different random seed a total of five times and report the average of those five different runs. We present these updated results below:
>
> |Model|Zero-Shot VAST |Few-Shot VAST |
> |-----------|-----------|-----------|
> | |Pro, Con, Neutral, All|Pro, Con, Neutral, All|
> |-----------|-----------|-----------|
> |DeBERTa|0.680, 0.683,0.900,0.755 | 0.659,0.657,0.869,0.728|
> |-----------|-----------|-----------|
> |TAG|0.681, 0.687,0.901,0.756 | 0.665,0.655,0.868,0.729|
> |-----------|-----------|-----------|
> |TAW|0.672, 0.709,0.903,0.760 | 0.656,0.677,0.869,0.736|
> |-----------|-----------|-----------|
> |TATA|0.695,0.711,0.905,0.771|0.665,0.683,0.873,0.741|
> |-----------|-----------|-----------|
>
> |Model|Qte |Sarc|Imp|mlS|mlT|
> |-----------|-----------|-----------|-----------|-----------|-----------|
> |DeBERTa|0.709|0.713|0.667|	0.578|0.676|
> |-----------|-----------|-----------|-----------|-----------|-----------|
> |TAG|0.710|0.731|0.668|	0.568|0.676|
> |-----------|-----------|-----------|-----------|-----------|-----------|
> |TAW|0.712|0.722|0.683|	0.585|0.693|
> |-----------|-----------|-----------|-----------|-----------|-----------|
> |TATA|0.714|0.730|0.693|	0.603|0.704|
> |-----------|-----------|-----------|-----------|-----------|-----------|
>
> We again observe that all of our models perform better than the previous state-of-the-art JointCl (0.723 in Zero-shot and 0.715 in the Few-shot) [1]. Furthermore, our TAG and TAW models perform better than our DeBERTa model in the Zero-shot setting evidencing that our approach *does* help for stance detection on unseen topics. As seen, we again see that our TATA model performs the best in both the Zero-shot and the Few-shot settings.
>
>
>
> ## References
>
>
> [1] Bin Liang, Qinlin Zhu, Xiang Li, Min Yang, Lin Gui, Yulan He, and Ruifeng Xu. 2022b. Jointcl: A joint contrastive learning framework for zero-shot stance detection. In Proceedings of the 60th Annual Meeting of the Association for Computational Linguistics (Volume 1: Long Papers), volume 1, pages 81–91. Association for Computational Linguistics.
>
> [2] Emily Allaway and Kathleen McKeown. 2020. Zero-shot stance detection: a dataset and model using
> generalized topic representations. arXiv preprint arXiv:2010.03640.
>
> [3] Nikunj Saunshi, Orestis Plevrakis, Sanjeev Arora, Mikhail Khodak, and Hrishikesh Khandeparkar.2019. A theoretical analysis of contrastive unsupervised representation learning. In International Conference on Machine Learning, pages 5628–5637. PMLR.
>
>
> [4] Hans WA Hanley and Zakir Durumeric. 2023. Machine-made media: Monitoring the mobilization of machine-generated articles on misinformation and mainstream news websites. arXiv preprint arXiv:2305.09820.

---

### Meta-Review · Area_Chair_s2yo · 2023-09-08

**Recommendation:** 3

**Metareview:**

This paper proposes a method for decoupling topic-agnostic and topic-aware representations for stance detection. This is done via two seperate embedding layers, trained via contrastive learning. The authors further introduce a new dataset for pre-training the topic-aware representations. This method leads to favourable empirical results. However, the method proposed is incremental and there are issues with the presentation. Some reviewers also ask for additional ablation results, which have been reported in the author response.

---

### Decision · Program_Chairs · 2023-10-07

**Decision:**

Accept-Main

**Comment:**

This paper proposes a method for decoupling topic-agnostic and topic-aware representations for stance detection. This is done via two seperate embedding layers, trained via contrastive learning. The authors further introduce a new dataset for pre-training the topic-aware representations. This method leads to favourable empirical results. However, the method proposed is incremental and there are issues with the presentation. Some reviewers also ask for additional ablation results, which have been reported in the author response.